# Identification of novel biomarkers to distinguish clear cell and non-clear cell renal cell carcinoma using bioinformatics and machine learning

Chanita Panwoon[1], Wunchana Seubwai[2,3], Malinee Thanee[1], Sakkarn Sangkhamanon[1] *

1 Faculty of Medicine, Department of Pathology, Khon Kaen University, Khon Kaen, Thailand, 2 Faculty of Medicine, Department of Forensic Medicine, Khon Kaen University, Khon Kaen, Thailand, 3 Faculty of Medicine, Center for Translational Medicine, Khon Kaen University, Khon Kaen, Thailand

* sakkarn@kku.ac.th

## Abstract

Renal cell carcinoma (RCC), accounting for 90% of all kidney cancer, is categorized into clear cell RCC (ccRCC) and non-clear cell RCC (non-ccRCC) for treatment based on the current NCCN Guidelines. Thus, the classification will be associated with therapeutic implications. This study aims to identify novel biomarkers to differentiate ccRCC from non-ccRCC using bioinformatics and machine learning. The gene expression profiles of ccRCC and non-ccRCC subtypes (including papillary RCC (pRCC) and chromophobe RCC (chRCC)), were obtained from TCGA. Differential expression genes (DEGs) were identified, and specific DEGs for ccRCC and non-ccRCC were explored using a Venn diagram. Gene Ontology and pathway enrichment analysis were performed using DAVID. The top ten expressed genes in ccRCC were then selected for machine learning analysis. Feature selection was operated to identify a minimum highly effective gene set for constructing a predictive model. The expression of best-performing gene set was validated on tissue samples from RCC patients using immunohistochemistry techniques. Subsequently, machine learning models for diagnosing RCC were developed using H-scores. There were 910, 415, and 835 genes significantly specific for DEGs in ccRCC, pRCC, and chRCC, respectively. Specific DEGs in ccRCC enriched in PD-1 signaling, immune system, and cytokine signaling in the immune system, whereas TCA cycle and respiratory, signaling by insulin receptor, and metabolism were enriched in chRCC. Feature selection based on Decision Tree Classifier revealed that the model with two genes, including NDUFA4L2 and DAT, had an accuracy of 98.89%. Supervised classification models based on H-score of NDUFA4L2, and DAT revealed that Decision Tree models showed the best performance with 82% accuracy and 0.9 AUC. NDUFA4L2 expression was associated with lymphovascular invasion, pathologic stage and pT stage in ccRCC. Using integrated bioinformatics and machine learning analysis, NDUFA4L2 and DAT were identified as novel biomarkers to differential diagnosis ccRCC from non-ccRCC.

**Data Availability Statement:** The transcriptomic data of three RCC subtypes are publicly available from OncoDB databases (https://oncodb.org/) [11]. The classification algorithms that are

performed in this study are available via a public GitHub repository (https://github.com/CHaPAN6551/machine-learning-classification-algorithm). The anonymized clinicopathological data from the study cohort is available in S7 Table.

**Funding:** The study was supported by grants from the Fundamental Fund of Khon Kaen University 2566, under the National Science, Research, and Innovation Fund or NSRF, Thailand. The funders had no role in study design, data collection and analysis, manuscript preparation, or publication decisions.

**Competing interests:** The authors affirm that there are no conflicts of interest, and all authors attest to the accuracy of the information presented.

## Introduction

Renal cell carcinoma (RCC) accounts for 90% of all kidney cancers. RCC has many recognizable subtypes listed in the 2022 WHO Blue Book [1,2]. Clear cell RCC (ccRCC) is the most common subtype, accounting for 75% of cases, while the remaining 25% is non-clear cell RCC (non-ccRCC) [3]. According to the NCCN Guidelines, managing patients with ccRCC and non-ccRCC significantly differs and relies on pathologic diagnosis. Therefore, the differential diagnosis of ccRCC from other types of RCC has become increasingly crucial [4]. Most RCCs can be diagnosed histologically by utilizing a limited panel of immunohistochemical studies. However, in some situations, particularly a high-grade or borderline case with overlapping morphology in which a pathologic diagnosis could not be made under routine histologic, even after performed immunohistochemistry, the result remains inconclusive. This setting necessitates sub-expertise pathologists who are specialized and experienced in this field. One of the issues the global community currently faces and will become increasingly concerned with is the worldwide shortage of pathologists, especially genitourinary pathologists (GU pathologists). Despite the promising approach of novel RCC management through radiogenomics, which offers the capability to characterize potentially malignant lesions in their genetic, epigenetic, and pathological heterogeneity via noninvasive methods such as advanced imaging study [5]. The comprehensive understanding of the differences in transcriptomic and proteomic profiles between ccRCC and non-ccRCC remains unclear. Therefore, understanding different molecule signatures and utilizing novel biomarkers along with bioinformatics and machine learning to differentiate ccRCC from non-ccRCC is an area of interest to explore.

Bioinformatics is a field that combines computational and experimental approaches in molecular and cell biology [6,7]. Machine learning is a branch of computer science that focuses on developing algorithms to learn from and make predictions or decisions based on data [8]. The integration of bioinformatics and machine learning techniques in cancer research is becoming more common. These methods greatly impact finding new molecular pathways, genetic changes, prognostic and diagnostic biomarkers, and efficient target molecules for different forms of cancer. Bioinformatics and machine learning enable the analysis of large-scale genomic, transcriptomic, and proteomic datasets by utilizing cutting-edge computational methodologies and data processing algorithms, facilitating the identification of crucial biomarkers and potential therapeutic targets in cancer research. In previous studies, data sets from The Cancer Genome Atlas (TCGA) were used to identify key genes as diagnostic genes and prognostic genes by integrating with classifier machine learning algorithms. Three classifiers random forest (RF), support vector machine (SVM), and neural network (NN) were used to perform classifier models to identify potential genes that distinguish the primary sites of metastatic carcinoma that metastasize to the cervical lymph node [9]. Likewise, genes associated with cancer development were pinpointed to assess the prognostic stage of clear cell renal cell carcinoma patients [10]. Both investigations underscore the power of machine learning and informatics in analyzing RNA sequencing data for cancer research, aiming to improve classification accuracy and shed light on molecular mechanisms and diagnostic approaches for different types of cancer. However, there are no reports on the use of integrated bioinformatics and machine learning techniques for determining novel biomarkers to differentiate ccRCC from non-ccRCC.

This study aimed to determine novel biomarkers to differentiate ccRCC from non-ccRCC. Differentially expressed genes (DEGs) in ccRCC and non-ccRCC, compared to normal tissue, were identified using a bioinformatics approach based on transcriptomics data from the TCGA database. The specific DEGs in ccRCC and non-ccRCC were identified using a Venn diagram. The recursive feature elimination (RFE) technique, based on the Decision Tree

Classifier, was used to identify the most important DEGs that can classify ccRCC from non-ccRCC. The expression levels of genes identified by RFE were confirmed in RCC patient tissues using immunohistochemistry techniques (IHC). Subsequently, classification models were generated based on the IHC score. Finally, the performance of these models was assessed using classification reports and ROC-AUC techniques.

## Materials & methods

### Datasets collection

The RNA expression profiles of common subtypes of RCC, namely clear cell RCC (KIRC) (S1 Table) and non-clear cell RCC (non-ccRCC), including papillary RCC (KIRP) (S2 Table), and chromophobe RCC (KICH) (S3 Table), were retrieved from TCGA (online publicly available database) through the OncoDB online database resource [11]. The RNA-Seq dataset for KIRC included 545 tumor samples and 72 normal samples, KIRP included 290 tumor samples and 32 normal samples, and KICH included 66 tumor samples and 25 normal samples.

### RCC patients and samples

The data and samples of RCC patients, including formalin-fixed paraffin-embedded (FFPE) tissue blocks and clinicopathologic information from January 1, 2017, to December 1, 2020, were first accessed from November 20, 2021, and completed accession on December 31, 2021. 58 FFPE tissues from those patients who underwent nephrectomy were collected from the sample archive of the Department of Pathology, Srinagarind Hospital, Faculty of Medicine, Khon Kaen University. Of these patients, 43 were diagnosed with ccRCC, while the remaining 15 were classified in the non-ccRCC group (including 5 cases of pRCC, 4 cases of chRCC, and 6 cases of other rare RCC subtypes). During the data collection step, the authors accessed information that could identify individual participants. However, in all experimental steps, the authors blinded the participants' identifiable information by replacing hospital numbers (HN) or identity-related data with index numbers (such as case No. 1–58).

   **Ethics statement.**   The study was conducted under the Declaration of Helsinki and the International Council for Harmonization (ICH) Good Clinical Practice Guidelines, and the protocol (HE641597) was approved by the Khon Kaen University Ethics Committee for Human Research. For the immunohistochemical study in tissue of patients, the consent form was not obtained due to the data being analyzed anonymously and its retrospective nature of data collection with less than minimal risk to the patient. The exemption of the consent form was approved by Khon Kaen University Ethics Committee for Human Research.

### Identification of Differentially Expressed Genes (DEGs) feature selection

The gene expression profiles of ccRCC and non-ccRCC were retrieved from TCGA using OncoDB (https://oncodb.org/). Genes with $|log2FC| \geq 1$ were identified as DEGs. The Venn diagram was used to identify genes that were specific to ccRCC and non-ccRCC. The intersection part of the Venn diagram was determined using JVENN [12].

### Gene Ontology (GO) and pathway enrichment analysis

GO and pathway enrichment analysis of specific DEGs were performed on the Database for Annotation, Visualization, and Integrated Discovery (DAVID) [13]. Enriched biological processes (BP), cellular components (CC), and molecular functions (MF) were identified with a significance threshold of $p$-value $< 0.05$ and FDR $< 0.05$. The GO and pathway enrichment analysis data were visualized into bubble plots using the ggplot2 R package [14].

## Feature selection

An essential step in developing machine learning models is using feature selection to identify the optimal set of features. This approach employs the fewest possible features to produce a predictive model. The top ten genes were initially selected in the present study based on the highest log2 fold change (log2FC). Subsequently, Recursive Feature Elimination (RFE) based on the DecisionTreeClassifier was utilized to select the minimal set of features necessary to generate models suitable for tissue analysis using the immunohistochemistry (IHC) technique.

## Machine learning model construction

The transcriptomics data were split into 70% and 30% for the training and test sets. The training set was performed on six classification algorithms, including Decision Tree (DT), Random Forest (RF), Logistic Regression (LR), K-nearest neighbors (KNN), Support Vector Machine (SVM), and Artificial Neural Network (ANN). Then, the performance of classification models was evaluated on a test set.

IHC data was performed on four classification algorithms, including Logistic Regression (LR), Decision Tree (DT), Random Forest (RF), and Gradient Booster (GB). However, the IHC data was imbalanced, including 43 ccRCC samples and 15 non-ccRCC samples. To solve the classes imbalance problem in the IHC data, the Synthetic Minority Oversampling Technique (SMOTE) was applied to oversampling the non-ccRCC group by increasing samples from 15 to 38 [15]. The data was split into 80% of the training and 20% of the test sets. The function "GridSearchCV" was used to tune the parameters of four algorithms to get the best performance.

Performance values such as receiver operating characteristic curves (ROC), area under the curve (AUC), accuracy, specificity, precision, recall, and f1 score values were used to evaluate the performance of each classifier in the test set.

## Immunohistochemistry staining

IHC was used to verify the expression levels of candidate genes from RFE and the classification model. The expression of candidate genes was determined in FFPE tissues from RCC patients. The FFPE blocks were sectioned with a microtome to a thickness of 4 micrometers and attached to glass slides. All samples were stained with NDUFA4L2 polyclonal antibody (dilution 1:500, Proteintech, Chicago, IL, USA) and DAT polyclonal antibody (dilution 1:100, Proteintech, Chicago, IL, USA). The detector used ultraView Universal DAB Detection Kit stainer (Ventana Medical Systems, Tucson, AZ, USA) that is an indirect, biotin-free system for detecting mouse IgG, mouse IgM and rabbit primary antibodies. The process was performed using Ventana Benchmark XT automated stainer (Ventana Medical Systems, Tucson, AZ, USA) according to the manufacturer's recommendations for visualization.

The expression levels of candidate genes were determined as Immunohistochemistry score (H-score) by a single pathologist. Ten randomly selected fields at 400 magnifications were used. The cytoplasmic and membranous staining intensity in malignant cells was graded on a scale of 0 to 3, where 0 represents negative staining, 1 represents weak staining, 2 represents moderate staining, and 3 represents strong brown staining. Cells were counted in each field, both overall and based on various staining intensities [16]. The average positive percentage was calculated using the following formula: H-score = (% of cells stained at intensity category 1 x 1) + (% of cells stained at intensity category 2 x 2) + (% of cells stained at intensity category 3 x 3). Genes expressed more than the cut-off value were interpreted as having high expression.

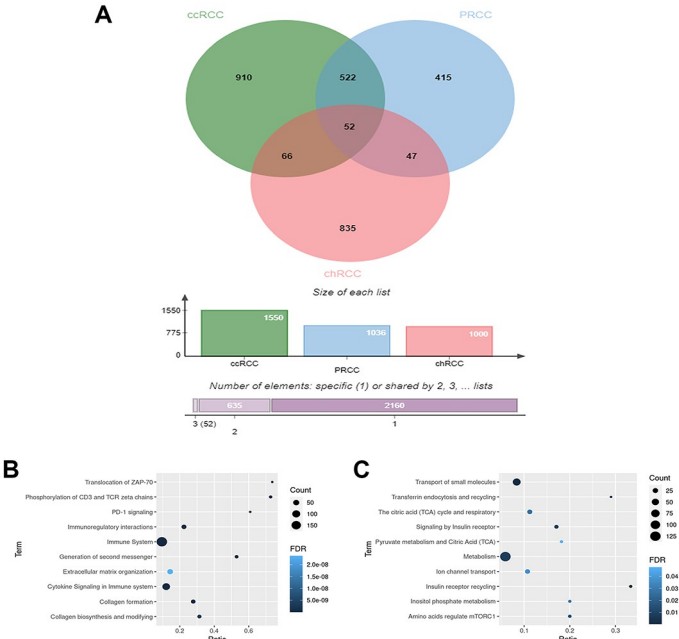

**Fig 1. Venn diagrams represent the overlaps of DEGs between three datasets of RCC subtypes and pathway enrichment analysis.** (A) Venn diagrams showing overlap of upregulated genes between ccRCC (green), pRCC (blue) and chRCC (red) dataset. (B and C) The bubble plots of the top 10 pathway enrichment for DEGs, B is the top 10 pathway enrichment of ccRCC, and C is the top 10 pathway enrichment of chRCC.

## Statistical analysis

Statistical analysis was performed using IBM SPSS Statistics v.28 (SPSS Inc., Chicago, IL). The Chi-square test ($x^2$-Test) and Fisher's exact test were used to calculate the correlations between the expression of five genes and clinicopathological characteristics, including age, sex, tumor size, tumor grade, lymphovascular invasion, and tumor stage. The performance of the biomarker panel was evaluated based on sensitivity and specificity. A $p$-value $< 0.05$ was set as statistical significance.

## Results

### Identification of DEGs

Based on the absolute log2FC, 1550, 1036, and 1000 DEGs were identified in ccRCC, pRCC, and chRCC, respectively. To investigate the specific DEGs in each RCC subtype, a Venn diagram was used. The results demonstrated that there were 910, 415, and 835 specific DEGs in ccRCC, pRCC, and chRCC, respectively (Fig 1A).

### GO and pathway enrichment analysis

GO and pathway enrichment analyses were performed to understand the functions and pathways of DEGs in each RCC subtype. The GO analysis data revealed that the specific DEGs of ccRCC were involved in the BP related to the immune system, particularly immune response, inflammatory response, T cell activation, and angiogenesis. The CC analyses showed correlations with the plasma membrane, cell surface, extracellular region, and MHC class II protein complex. In terms of MF, these DEGs were associated with identical protein binding, MHC

class II receptor activity, and transmembrane signaling receptor activity (S4 Table). While, in pRCC subgroups, these DEGS were associated with cartilage development, extracellular space, and receptor binding (S5 Table), for chRCC, the specific DEGS were involved in hydrogen ion transmembrane transport, an integral component of membranes, and actin binding (S6 Table).

Pathway enrichment analyses in three RCC subtypes revealed that the DEGs of ccRCC were significant for the immune system, cytokine signaling in the immune system, and phosphorylation of CD3 and TCR zeta chains (Fig 1B). In contrast, the DEGs in chRCC were involved in metabolic pathways such as the transport of small molecules, insulin receptor recycling, and metabolism (Fig 1C). On the other hand, DEGs of pRCC did not show a significant relationship with any specific pathway.

## Feature selection

The top ten DEGs in ccRCC with high log2FC, including NDUFA4L2, FABP7, DAT, ANGPTL4, NPTX2, CYP2J2, EGNL3, CP, SEMA58, and CDCA2 (Table 1), were selected to perform feature selection using RFE technique based on Decision Tree Classifier. Different numbers of genes (1, 2, 3, 5, and 10) were assessed in the feature selection procedure.

The results of the RFE indicated that a set of five genes achieved the best performance, with an accuracy of 99.5% (Table 2). However, considering health economics, particularly the costs associated with IHC staining, using a smaller number of biomarkers while maintaining high accuracy might be the most practical option for real-world application. In this context, using only 2 genes, namely NDUFA4L2 and DAT, had 98.9%, which achieved an accuracy of 98.9%, comparable to that of the 5 genes. Therefore, these two genes were selected as features for constructing the classification models.

Six classifier algorithms (LR, DT, RF, KNN, SNM, and ANN) were used to generate a classification model for differentiating ccRCC and non-ccRCC. The classification model's performance was evaluated using accuracy, classification report, and ROC-AUC plot (Table 3 and Fig 2). The results found that all models demonstrated accuracy ranging from 96% to 98%, specificity ranging from 97% to 99%, sensitivity ranging from 96% to 97%, and AUC values ranging from 96% to 100% (Table 3). The ANN models exhibited the best performance, achieving 97.8% accuracy, 99% specificity, 97.1% sensitivity, and 99.8% AUC. This was closely followed by the LR model, which showed 97.1% accuracy, 99% specificity, 95.9% sensitivity, and 99.8% AUC.

**Table 1. The top ten genes were selected based on FDR and |log2FC|.**

| Genes | |log2FC| | FDR-Adjusted | Genes name |
|---|---|---|---|
| NDUFA4L2 | 7.02 | q< = 5.7e-20 | NDUFA4 Mitochondrial Complex Associated Like 2 |
| FABP7 | 6.77 | q< = 5.7e-20 | Fatty Acid Binding Protein 7 |
| DAT | 6.52 | q< = 5.7e-20 | Dopamine Transporter |
| ANGPTL4 | 6.1 | q< = 5.7e-20 | Angiopoietin Like 4 |
| NPTX2 | 5.96 | q< = 5.7e-20 | Neuronal Pentraxin 2 |
| CYP2J2 | 5.23 | q< = 5.7e-20 | Cytochrome P450 Family 2 Subfamily J Member 2 |
| EGLN3 | 4.44 | q< = 5.7e-20 | Egl-9 Family Hypoxia Inducible Factor 3 |
| CP | 4.28 | q< = 5.7e-20 | Ceruloplasmin |
| SEMA5B | 4.03 | q< = 5.7e-20 | Semaphorin 5B |
| CDCA2 | 3.92 | q< = 5.7e-20 | Cell Division Cycle Associated 2 |

**Table 2. The performance of different feature sets in the feature selection process by using the Decision Tree Classifier.**

| Features | Accuracy (%) | Precision (%) | | Recall (%) | | f1-score (%) | |
|---|---|---|---|---|---|---|---|
| | | ccRCC | Non-ccRCC | ccRCC | Non-ccRCC | ccRCC | Non-ccRCC |
| 10 | 98.9 | 99.1 | 98.6 | 99.1 | 98.6 | 99.1 | 98.6 |
| 5 | 99.5 | 99.1 | 100.0 | 100.0 | 98.6 | 99.6 | 99.3 |
| 3 | 97.8 | 97.4 | 98.5 | 99.1 | 95.7 | 98.3 | 97.1 |
| 2 | 98.9 | 100.0 | 97.2 | 98.2 | 100.0 | 99.1 | 98.6 |
| 1 | 90.1 | 92.7 | 85.9 | 91.1 | 88.4 | 91.9 | 87.1 |

## Validation of NDUFA4L2 and DAT expression in RCC patient tissues

The expression of NDUFA4L2 and DAT were validated in 43 tissue samples of ccRCC and 15 tissue samples of non-ccRCC using the IHC technique. Both NDUFA4L2 and DAT were expressed in membranous and cytoplasmic patterns. To determine these genes' expression levels, the H-score median was used as a cut-off value. NDUFA4L2 and DAT were expressed in both ccRCC and non-ccRCC. The expression of NDUFA4L2 and DAT in both groups, ccRCC and non-ccRCC, was not statistically different (Table 4). The mean expression score of NDUFA4L2 in the ccRCC and non-ccRCC were 200.1 (SD = 50.7, median = 207.5) and 220.1 (SD = 41.4, median = 230.5), respectively (Fig 3A). Similarly, the expression scores of DAT in ccRCC and non-ccRCC were 128.1 (SD = 51.5, median = 122) and 168.8 (SD = 46.4, median = 180.5), respectively (Fig 3B). Therefore, the cases with expression levels higher than the cut-off value (median of H-score) would be classified as high expression in both proteins. An example of the IHC study is shown in Figs 4 and 5.

## The correlation of NDUFA4L2 and DAT expression and clinicopathological feature

The correlation between NDUFA4L2/DAT expression and clinicopathological features of RCC patients was investigated. In the ccRCC group, NDUFA4L2 expression was significantly related to lymphovascular invasion ($p$ = 0.043), prognosis stage ($p$ = 0.023), and pT stage ($p$ = 0.022) (Table 5). However, in the non-ccRCC group, NDUFA4L2 expression was not significantly correlated with any clinicopathological features. In addition, the expression of DAT demonstrated no significant difference in any clinicopathological features in both groups (Table 6).

**Table 3. The performance of six classifier models to distinguish between ccRCC and non-ccRCC on a test set based on transcriptomic datasets.**

| Models | Spec. (%) | Acc. (%) | AUC (%) | Precision (%) | | Recall (%) | | F1-score (%) | |
|---|---|---|---|---|---|---|---|---|---|
| | | | | ccRCC | Non-ccRCC | ccRCC | Non-ccRCC | ccRCC | Non-ccRCC |
| DT | 97.0 | 96.7 | 96.7 | 98.2 | 94.2 | 96.5 | 97.0 | 97.4 | 95.6 |
| RF | 98.0 | 97.8 | 98.9 | 98.8 | 96.1 | 97.7 | 98.0 | 98.2 | 97.0 |
| LR | 99.0 | 97.1 | 99.8 | 99.4 | 93.4 | 95.9 | 99.0 | 97.6 | 96.1 |
| KNN | 98.0 | 97.8 | 99.3 | 98.8 | 96.1 | 97.7 | 98.0 | 98.2 | 97.0 |
| SVM | 98.0 | 97.8 | 99.5 | 98.8 | 96.1 | 97.7 | 98.0 | 98.2 | 97.0 |
| ANN | 99.0 | 97.8 | 99.8 | 99.4 | 95.2 | 97.1 | 99.0 | 98.2 | 97.1 |

In statistical terms, recall of ccRCC is equal to sensitivity. DT: Decision tree, RF: Random forest, LR: Logistic regression, KNN: K-nearest neighbor, SVM: Support vector machine, ANN: Artificial neural network, Spec.: Specificity, Acc.: Accuracy, AUC: Area under the curve.

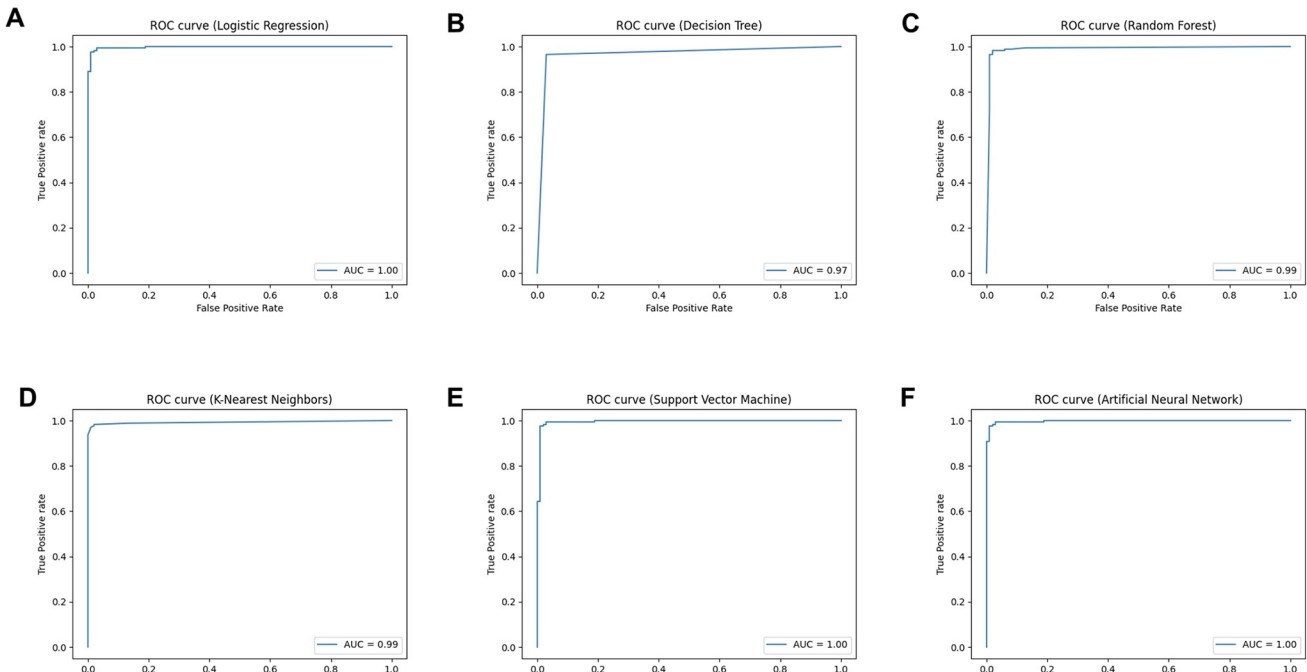

**Fig 2. The ROC curve and AUC score of six predictive models for determining ccRCC and non-ccRCC based on transcriptomic datasets.** (A-F) The ROC curves of LR, DT, RF, KNN, SVM and ANN, respectively. The x-axis represents a false positive rate, and the y-axis represents a true positive rate. Logistic Regression (LR), Decision Tree (DT), Random Forest (RF), K-Nearest Neighbors (KNN), Support Vector Machine (SVM) and Artificial Neural Network (ANN).

## The classification model based on IHC scores

We next perform classification models using the IHC data of NDUFA4L2 and DAT. Four supervised classification algorithms, including DT, RF, LR, and GB, were used in the present study. The over-resampling technique using Synthetic Minority Over-sampling Technique (SMOTE) [15] was used to solve the class imbalance problem in the IHC dataset. The DT models showed the best performance with 82.4% accuracy, 100% specificity, 70% sensitivity and 90% AUC. Followed by GB models had 82.4% accuracy, 85.7% specificity, 80% sensitivity, and 83% AUC. For other models, the RF model had 70.6% accuracy, 85.7% specificity, 60% sensitivity, and 73% AUC. For the LR model had 52.9% accuracy, 85.7 specificity, 30% sensitivity, and 57% AUC (Table 7 and Fig 6).

**Table 4. The Correlation of two gene expressions with RCC groups by IHC technique.**

| Gene expression | | Clear cell RCC | Non-clear cell RCC | p-value |
|---|---|---|---|---|
| | | n (%) | n (%) | |
| NDUFA4L2 | Low | 20 (46.5) | 7 (46.7) | 0.992 |
| | High | 23 (53.6) | 8 (53.3) | |
| DAT | Low | 21 (48.8) | 7 (46.7) | 0.885 |
| | High | 22 (51.2) | 8 (53.3) | |

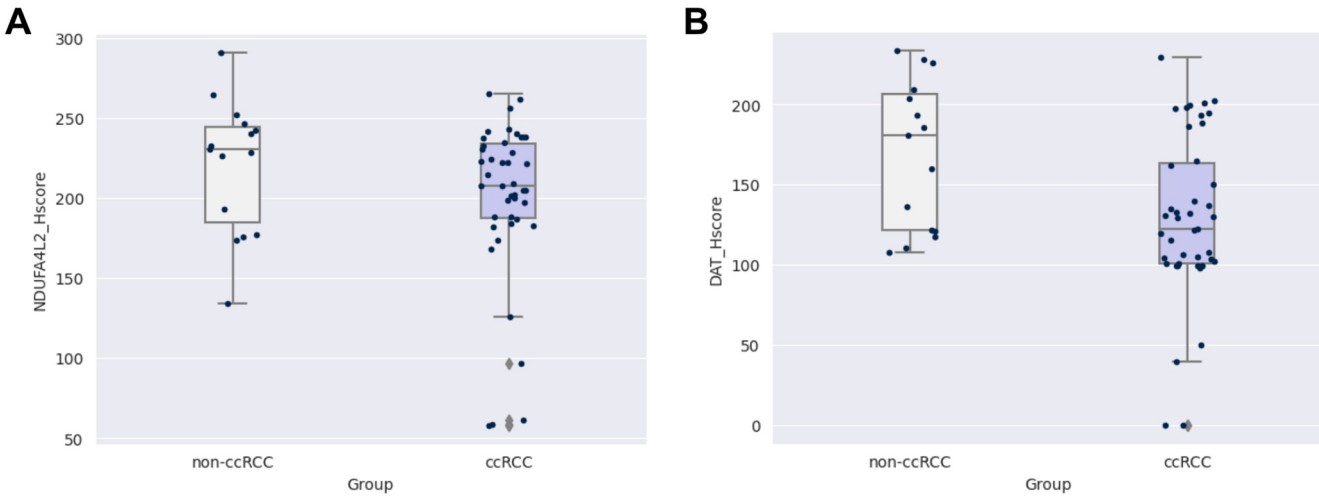

**Fig 3. The box plot diagrams represent the correlations between gene expression distribution of NDUFA4L2 and DAT with RCC groupings.** (A) The box plot of NDUFA4L2 expression between ccRCC and non-ccRCC grouping. (B) The box plot of DAT expression between ccRCC and non-ccRCC grouping. The x-axis represents the groups of RCC, and the y-axis represents the gene expression values based on H-score.

## Discussion

RCC is suggested as a metabolic disease, with alterations in various pathways like energy metabolism, fatty acid metabolism, and amino acid processing [17,18]. Wettersten et al. suggest a reprogramming of sugar and fat usage, along with changes in the tricarboxylic acid cycle, often linked to mutations in the VHL gene [18]. Lucarelli et al. reported that a high level of glucose-6-phosphate dehydrogenase (G6PD) because of increasing pentose phosphate pathway (PPP)-derived metabolites led to enhanced activity in both glycolysis and the PPP in ccRCC [19]. These findings suggest energy metabolic and glycolysis reprogramming is key in promoting RCC. Additionally, Schaeffeler and co-workers explored that RCC subtypes had different biological and metabolic profiles based on the correlation of the original region [17].

In the era of precision medicine, artificial intelligence (AI) and machine learning (ML) have been applied using a multi-omics approach to enhance performance in detecting and differentiating urological tumors. Particularly successful were radiography studies for kidney and bladder lesions [20,21]. Integrating radiologic diagnosis with AI and ML is currently undergoing widespread development. However, there are limited studies in the pathology field. Thus, there is an opportunity to integrate transcriptomic analysis and ML that can be applied in pathology practice. Several transcriptomics datasets can be accessed via public databases such as TCGA and Gene Expression Omnibus (GEO). Integrative analysis using bioinformatics and machine learning (ML) techniques based on transcriptomics are useful for various medical purposes such as the determination of novel biomarkers for cancer prognosis and diagnosis [9,22], and the discovery of targeted drugs [23]. The application of integrative analysis through bioinformatics and ML enhances efficacy and significantly reduces both time and associated costs.

In this study, integrative analysis using bioinformatics ML technique based on transcriptomics of RCCs was conducted to identify novel biomarkers that can split two major subgroups of RCCs (ccRCC vs non-ccRCC). Pathway enrichment analyses showed that the DEGs of ccRCC were significant for the immune system. In contrast, the DEGs in chRCC were involved in metabolic pathways, while pRCC was not significantly associated with any

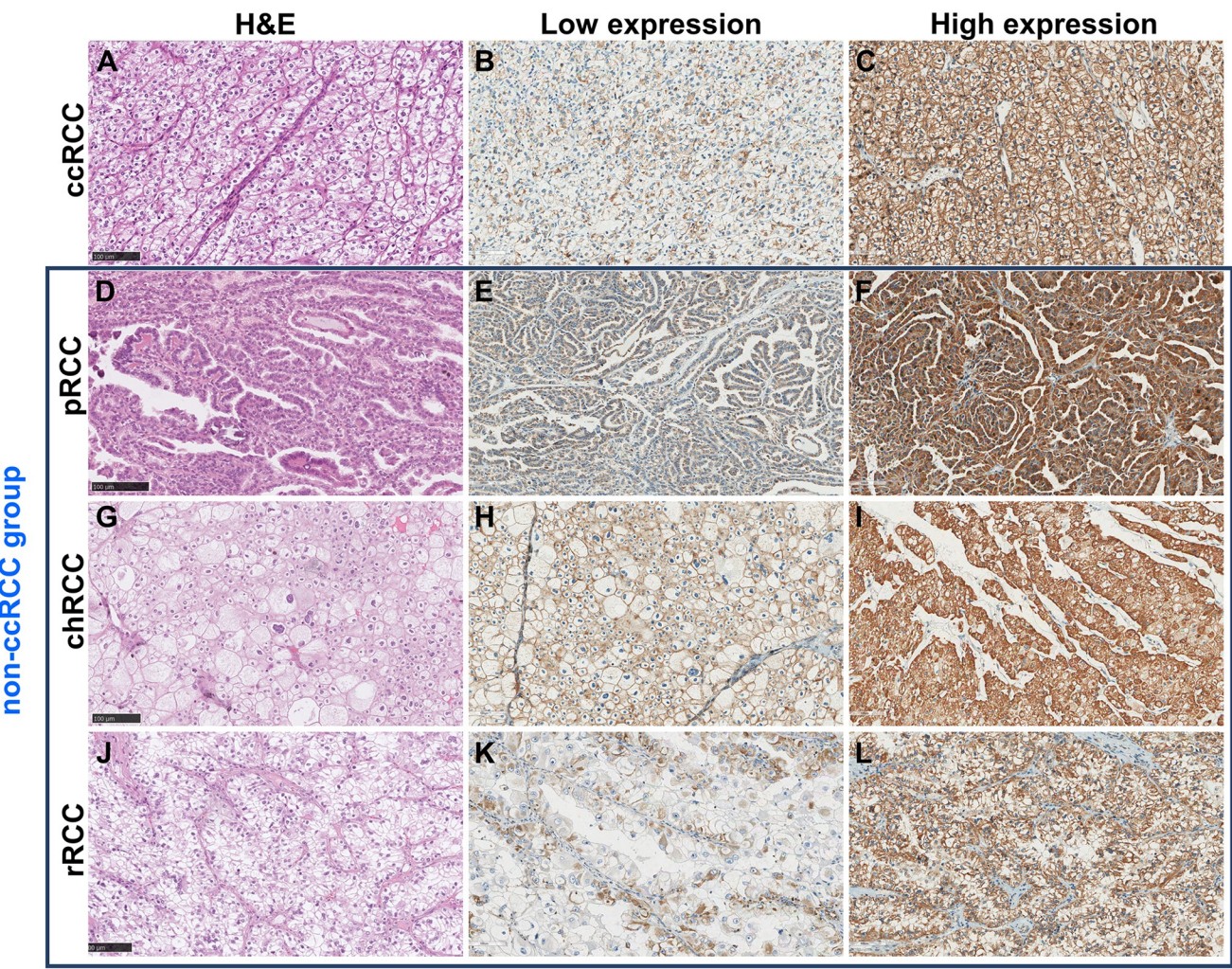

**Fig 4. Expression of NDUFA4L2 on tissues in ccRCC and non-ccRCC group.** (A-C) These demonstrated ccRCC group, (D-F) pRCC subtypes, (G-I) chRCC subtypes, and (J-L) rRCC subtypes (the examples of cases with histo- and immunologically suspected translocation-type RCC). The H&E staining of each histologic type including ccRCC (A), pRCC (D), chRCC (G), and rRCC (J) subtypes. The representative of low NDUFA4L2 expression in each subtype was demonstrated (B, E, H and K). The high expression of NDUFA4L2 in each subtype was shown (C, F, I and L). clear cell renal cell carcinoma (ccRCC), papillary renal cell carcinoma (pRCC), chromophobe renal cell carcinoma (chRCC) and rare subtypes renal cell carcinoma (rRCC). (All figures were taken at 20X).

pathways. Dysregulated key immune-associated genes and pathways were reported in ccRCC, suggesting potential immunotherapeutic targets in ccRCC patients [24]. These findings highlight the well-known fact of molecular heterogeneity between different subtypes of RCCs. Understanding these distinct molecular pathways is crucial for developing personalized therapies in different RCC subtypes.

Two potential biomarkers, namely NDUFA4L2 and DAT, were selected by the RFE feature selection algorithm. Various supervised ML algorithms, including LR, DT, RF, KNN, SNM and ANN, were employed to develop classification models. These ML models aimed at diagnosing ccRCC and non-ccRCC based on the mRNA expression values of NDUFA4L2 and DAT from TCGA dataset. The results indicated that all ML models demonstrated high performance with 96–98% accuracy, 97–99% specificity, 96–97% sensitivity, and 96–100% AUC.

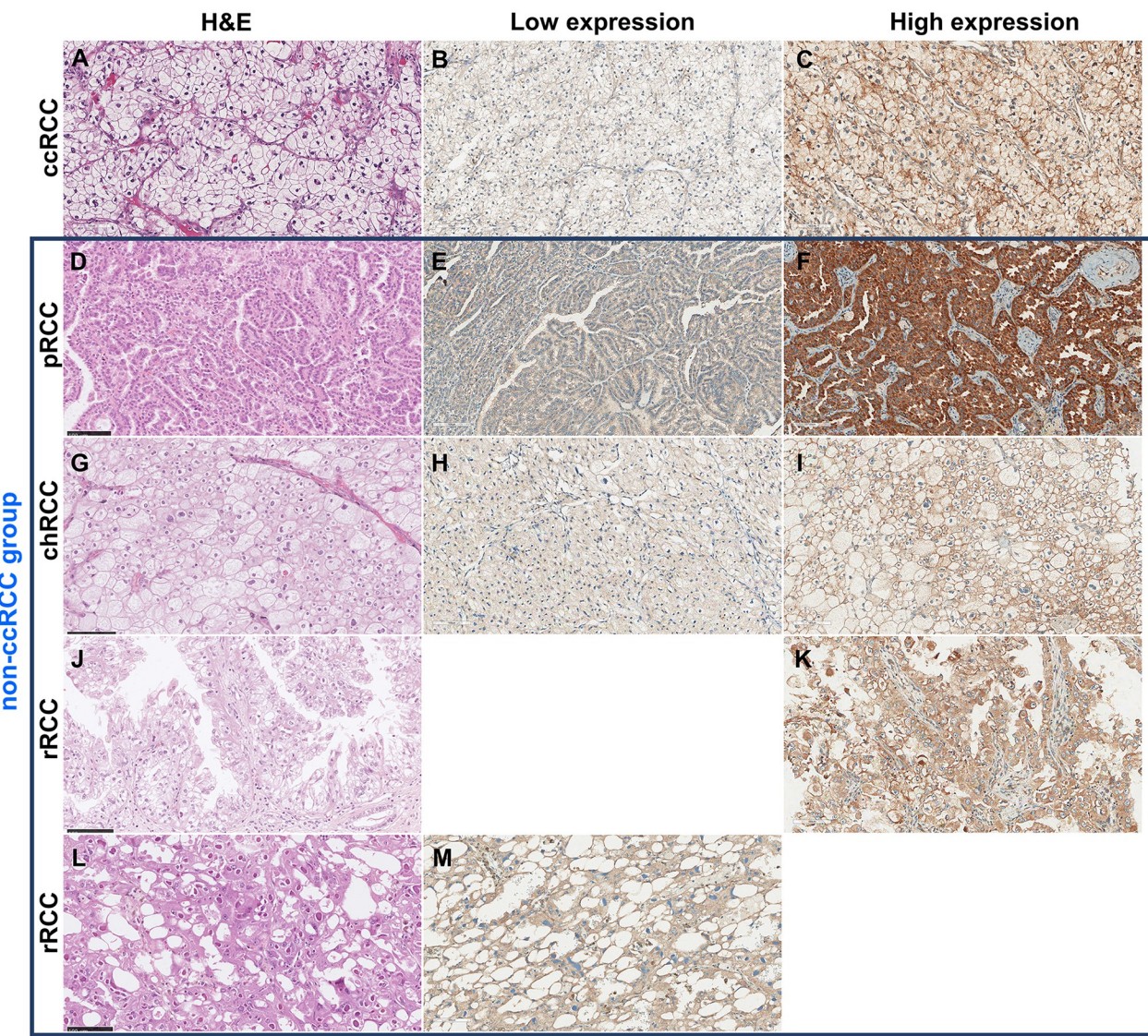

**Fig 5. Expression of DAT on tissues in ccRCC and non-ccRCC group.** (A-C) These demonstrated ccRCC group, (D-F) pRCC subtypes, (G-I) chRCC subtypes and (J-M) rRCC subtypes (J and K demonstrated that cases with histo- and immunologically suspected translocation-type RCC. L and M represented cases with histo- and immunologically suspected acquired cystic disease associated renal cell carcinoma). The H&E staining of each histologic type including ccRCC (A), pRCC (D), chRCC(G) and rRCC (J). The low expression of DAT was shown in ccRCC (B), pRCC (E), chRCC (H) and rRCC (M). The high signal of DAT staining in each subtype was shown in ccRCC (C), pRCC (F), chRCC (I) and rRCC (K). clear cell renal cell carcinoma (ccRCC), papillary renal cell carcinoma (pRCC), chromophobe renal cell carcinoma (chRCC) and rare subtypes renal cell carcinoma (rRCC). (All figures were taken at 20X).

In this study, the expression values of NDUFA4L2 and DAT were validated in FFPE from RCC patients using IHC technique. However, the IHC staining result was not according to the transcriptomics data. The expression of NDUFA4L2 and DAT in tissues from 43 ccRCCs and 15 non-ccRCCs were not significantly different. Similar findings were demonstrated in several reports. Gry et al. studied 23 human cell lines from multi-original cancer, found that the correlation coefficients between levels of RNA and protein products of specific genes varied widely [25]. Several factors affect the concordance between gene and protein expression such as

**Table 5. The correlation of NDUFA4L2 expression and clinicopathological features.**

| Clinicopathology features | ccRCC group | | | | non-ccRCC group | | | |
|---|---|---|---|---|---|---|---|---|
| | No. n (%) | NDUFA4L2 Expression | | p-value | No. n (%) | NDUFA4L2 Expression | | p-value |
| | | Low | High | | | Low | High | |
| **Sex** | | | | 0.801 | | | | 0.619 |
| Male | 33 | 15 (45.5) | 18 (54.5) | | 8 | 3 (37.5) | 5 (62.5) | |
| Female | 10 | 5 (50.0) | 5 (50.0) | | 7 | 4 (57.1) | 3 (42.9) | |
| **Age** | | | | 0.920 | | | | 1.000 |
| <55 | 19 | 9 (47.4) | 10 (52.6) | | 10 | 5 (50.0) | 5 (50.0) | |
| ≥55 | 24 | 11 (45.8) | 13 (54.2) | | 5 | 2 (40.0) | 3 (60.0) | |
| **WHO/ISUP Grade** | | | | 0.739 | | | | 1.000 |
| Low grade (I-II) | 29 | 14 (48.3) | 15 (51.7) | | 3 | 2 (33.3) | 1 (66.7) | |
| High grade (III-IV) | 14 | 6 (42.9) | 8 (57.1) | | 7 | 3 (42.9) | 4 (57.1) | |
| **Tumor size** | | | | 0.750 | | | | 1.000 |
| ≤7 | 29 | 13 (44.8) | 16 (55.2) | | 8 | 4 (50.0) | 4 (50.0) | |
| >7 | 14 | 7 (50.0) | 7 (50.0) | | 7 | 3 (42.9) | 4 (57.1) | |
| **Lymphovascular Invasion** | | | | 0.043* | | | | 1.000 |
| Absent | 30 | 17 (56.7) | 13 (43.3) | | 10 | 5 (50.0) | 5 (50.0) | |
| Present | 13 | 3 (23.1) | 10 (76.9) | | 5 | 3 (60.0) | 2 (40.0) | |
| **Prognostic stage** (TNM stage) | | | | 0.023* | | | | 1.000 |
| Early stage | 26 | 16 (61.5) | 10 (38.5) | | 11 | 5 (45.5) | 6 (54.5) | |
| Late stage | 13 | 3 (23.1) | 10 (76.9) | | 4 | 2 (50.0) | 2 (50.0) | |
| **pT stage** | | | | 0.022* | | | | 1.000 |
| Low-pT (pT1, pT2) | 29 | 17 (58.6) | 12 (41.4) | | 11 | 5 (45.5) | 6 (54.5) | |
| High-pT (pT3, pT4) | 14 | 3 (21.4) | 11 (78.6) | | 4 | 2 (50.0) | 2 (50.0) | |
| **Sarcomatoid** | | | | 0.365 | | | | 1.000 |
| Absent | 39 | 19 (48.7) | 20 (51.3) | | 8 | 4 (50.0) | 4 (50.0) | |
| Present | 4 | 1 (25.0) | 3 (75.0) | | 2 | 1 (50.0) | 1 (50.0) | |
| **Rhabdoid** | | | | 0.485 | | | | 1.000 |
| Absent | 37 | 18(48.6) | 19 (51.4) | | 5 | 3 (60.0) | 2 (40.0) | |
| Present | 6 | 2 (33.3) | 4 (66.7) | | 5 | 2 (40.0) | 3 (60.0) | |

* $p < 0.05$ versus genes expression values and clinicopathological feature in each group.

protein expression levels, technological sensitivities, sample processing, probe set definition or anatomical origin of tissue, and actual biological differences between transcript and protein abundance [26]. Subsequently, additional classification models were performed using 4 supervised ML algorithms, including LR, DT, RF and GB, based on IHC data of NDUFA4L2 and DAT in RCC tissues. It was found that the performance of the four models was satisfactory, especially the DT model (82.4% accuracy and 90% AUC) and the GB model (82.4% accuracy and 83% AUC).

This study found that NDUFA4L2 expression was correlated with lymphovascular invasion, pathologic stage (pT), and prognostic stage (TNM stage). Similarly, previous studies reported that NDUFA4L2 mRNA and protein were significantly increased in ccRCC compared with normal renal tissues [27,28], which is also associated with high-grade tumors. Furthermore, NDUFA4L2, a direct target gene of HIF-1α [27], contributes to the Warburg effect [29]. The high expression of NDUFA4L2 regulates glucose and energy metabolism alteration in cancer cells, promoting increased glucose uptake, a shift towards glycolysis, and altered mitochondrial

**Table 6. The correlation of DAT expression and clinicopathological features.**

| Clinicopathology features | ccRCC group | | | | non-ccRCC group | | | |
|---|---|---|---|---|---|---|---|---|
| | No. n (%) | DAT Expression | | *p*-value | No. n (%) | DAT Expression | | *p*-value |
| | | Low | High | | | Low | High | |
| **Sex** | | | | 0.523 | | | | 0.619 |
| Male | 33 | 17 (51.5) | 16 (48.5) | | 8 | 3 (37.5) | 5 (62.5) | |
| Female | 10 | 4 (40.0) | 6 (60.0) | | 7 | 4 (57.1) | 3 (42.9) | |
| **Age** | | | | 0.864 | | | | 0.608 |
| <55 | 19 | 9 (47.4) | 10 (52.6) | | 10 | 4 (40.0) | 6 (60.0) | |
| ≥55 | 24 | 12 (50.0) | 12 (50.0) | | 5 | 3 (60.0) | 2 (40.0) | |
| **WHO/ISUP Grade** | | | | 0.449 | | | | 0.183 |
| Low grade (I-II) | 29 | 13 (44.8) | 16 (55.2) | | 3 | 2 (33.3) | 1 (66.7) | |
| High grade (III-IV) | 14 | 8 (57.1) | 6 (42.9) | | 7 | 1 (14.3) | 6 (85.7) | |
| **Tumor size** | | | | 0.916 | | | | 0.619 |
| ≤7 | 29 | 14 (48.3) | 15 (51.7) | | 8 | 3 (37.5) | 5 (62.5) | |
| >7 | 14 | 7 (50.0) | 7 (50.0) | | 7 | 4 (57.1) | 3 (42.9) | |
| **Lymphovascular Invasion** | | | | 0.665 | | | | 0.282 |
| Absent | 30 | 14 (46.7) | 16 (53.3) | | 10 | 6 (60.0) | 4 (40.0) | |
| Present | 13 | 7 (53.8) | 6 (46.2) | | 5 | 1 (20.0) | 4 (80.0) | |
| **Prognostic stage** (TNM stage) | | | | 0.821 | | | | 0.569 |
| Early stage | 26 | 13 (50.0) | 13 (50.0) | | 11 | 6 (54.5) | 5 (45.5) | |
| Late stage | 13 | 7 (53.8) | 6 (46.2) | | 4 | 1 (25.0) | 3 (75.0) | |
| **pT stage** | | | | 0.916 | | | | 0.569 |
| Low-pT (pT1, pT2) | 29 | 14 (48.3) | 15 (51.7) | | 11 | 6 (54.5) | 5 (45.5) | |
| High-pT (pT3, pT4) | 14 | 7 (50.0) | 7 (50.0) | | 4 | 1 (25.0) | 3 (75.0) | |
| **Sarcomatoid** | | | | 0.317 | | | | 1.000 |
| Absent | 39 | 20 (51.3) | 19 (48.7) | | 8 | 2 (25.0) | 6 (75.0) | |
| Present | 4 | 1 (25.0) | 3 (75.0) | | 2 | 1 (50.0) | 1 (50.0) | |
| **Rhabdoid** | | | | 0.951 | | | | 1.000 |
| Absent | 37 | 18 (48.6) | 19 (51.4) | | 5 | 2 (40.0) | 3 (60.0) | |
| Present | 6 | 3 (50.0) | 3 (50.0) | | 5 | 1 (20.0) | 4 (80.0) | |

* *p* < 0.05 versus genes expression values and clinicopathological feature in each group.

**Table 7. The performance of two markers was evaluated on four classification models for separating ccRCC from non-ccRCC based on the H-score (immunohistochemistry score).**

| Models | Spec. (%) | Acc. (%) | AUC (%) | Precision (%) | | Recall (%) | | F1-score (%) | |
|---|---|---|---|---|---|---|---|---|---|
| | | | | ccRCC | Non-ccRCC | ccRCC | Non-ccRCC | ccRCC | Non-ccRCC |
| DT | 100.0 | 82.4 | 90.0 | 100.0 | 70.0 | 70.0 | 100.0 | 82.0 | 82.0 |
| RF | 85.7 | 70.6 | 73.0 | 86.0 | 60.0 | 60.0 | 86.0 | 71.0 | 71.0 |
| LR | 85.7 | 52.9 | 57.0 | 75.0 | 46.0 | 30.0 | 86.0 | 43.0 | 76.0 |
| GB | 85.7 | 82.4 | 83.0 | 89.0 | 75.0 | 80.0 | 86.0 | 84.0 | 80.0 |

In statistical terms, recall of ccRCC is equal to sensitivity. DT: Decision tree, RF: Random forest, LR: Logistic regression, KNN: K-nearest neighbor, SVM: Support vector machine, ANN: Artificial neural network, Spec.: Specificity, Acc.: Accuracy, AUC: Area under the curve.

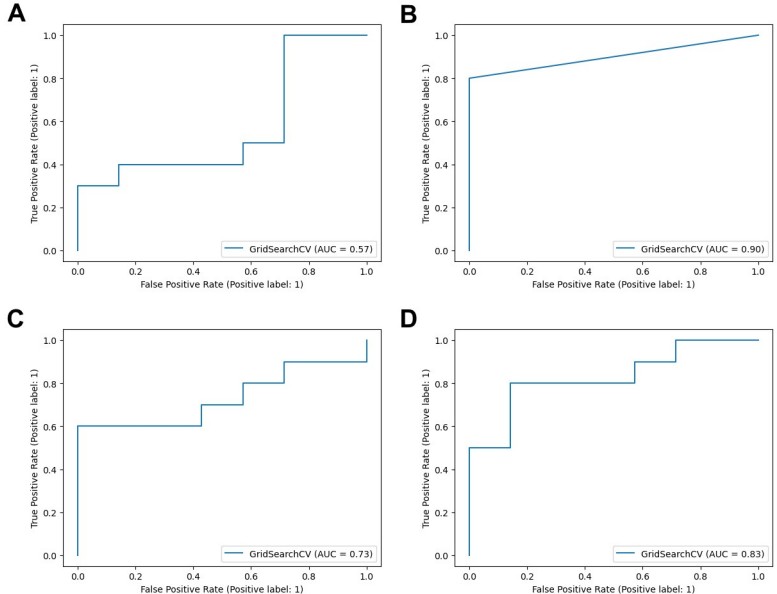

**Fig 6. The ROC cure and AUC score of four predictive models on IHC dataset.** (A-D) The ROC curves of LR, DT, RF and GB, respectively. The x-axis represents a false positive rate, and the y-axis represents a true positive rate.

function [28]. NDUFA4L2 was also involved in tumorigenesis pathways, such as insulin-like growth factor 1 (IGF-1), mammalian target of Rapamycin (mTOR), and phosphoinositide 3 kinase serine/threonine protein kinase (PI3K/AKT) [30], aerobic glycolysis, reduced oxygen consumption, and lowered ROS production [31]. The high expression of NDUFA4L2 was associated with poor prognostic outcomes in ccRCCs due to enhanced cell proliferation and anti-apoptosis. Conversely, NDUFA4L2 silencing disrupts cellular processes, including an inhibition of the autophagic machine, increased mitochondrial mass, and induced an overproduction of ROS, especially in hypoxic conditions [28]. NDUFA4L2 knockdown also had an antiproliferation effect in ccRCC cell culture [27]. Likewise, NDUFA4L2 protein knockdown in RCC decreased their viability and made them more susceptible to cisplatin chemotherapy. This suggested NDUFA4L2 plays a role in the resistance of chemotherapy of RCC. In particular, ccRCC stands out as a tumor with significant immune infiltration. This infiltration in tumors has been linked to the VHL tumor suppressor gene, triggering HIF-driven cellular responses, angiogenesis, and metabolic alterations [32]. Furthermore, previous study has revealed that metabolic-associated genes like MUC1 can influence immunoflogosis within the ccRCC microenvironment by activating the classical pathway of the complement system and modulating immune infiltrates, thereby fostering an immune-silent microenvironment [33]. These findings displayed the hidden significance of NDUFA4L2 in potentially inducing modulation of immune cell infiltration and regulating immunoflogosis by controlling metabolic alterations in ccRCC. In summary, the evidence suggests that targeting NDUFA4L2 could be a potential immunotherapeutic target of ccRCC. The NDUFA4L2 mutation also plays a role in multiple types of cancer, such as trastuzumab-resistant (TR) HER2-positive breast cancer [31], Non-small cell lung cancer (NSCLC) [34], Glioblastoma (GBM) [35], Hepatocellular carcinoma (HCC) [36] and colorectal cancer [37]. Possibly, NDUFA4L2 might be a potential therapeutic target in many cancers.

DAT, or the dopamine transporter gene (SLC6A3), is a member of the solute carriers (SLC) family, one of two major transporter proteins. From this investigation, DAT also had a high expression level in ccRCC compared with non-ccRCC groups based on RNASeq. In contrast, the result of the IHC technique demonstrated no statistical differential expression between ccRCC and non-ccRCC. Further, clinicopathological data showed no significant differences in each feature. Previous studies suggest that DAT was expected to be a specific biomarker of prognosis or targeted therapy for ccRCC. Hansson J et al. reported that DAT showed high expression transcript levels in ccRCC compared to normal tissue and other tumor types based on in silico analysis. Moreover, DAT was regulated by HIF-2α and induced by hypoxia in normal renal cells [38]. Contrarily, another study found that mRNA and protein expression of DAT were decreased in ccRCC compared with normal kidney; however, DAT protein was significantly increased in patients with distant metastasis but was not correlated with AJCC stage, pT-stage, lymph node metastasis, or tumor grading. It was also associated with shorter overall survival. Additionally, they found that treatment with the SLC6A3 inhibitor sertraline inhibited cell proliferation in the ccRCC cell line [39]. Similarly, DAT had a high expression level in gastric cancer and was reduced in the serum of patients with gastric cancer after surgery. Therefore, DAT might be a serological diagnostic marker of gastric cancer [40]. Nevertheless, the expression and role of the DAT gene in RCC are not widely studied, leading to a lack of understanding of its mechanisms. Therefore, future research may focus more on investigating the DAT gene in RCC.

This study faced significant limitations due to the small sample size and class imbalance issues among the RCC patient samples. These challenges could potentially impact the reliability of our findings. The larger sample size should further validate the expression of these genes. The investigation of functional studies on important genes in *in vitro* models might be used to be a potential target for RCC treatment in the future.

## Conclusions

NDUFA4L2 and DAT were identified as potential biomarkers for classifying RCC subtypes (ccRCC and non-ccRCC). The supervised ML models based on RNA-Seq and IHC data of NDUFA4L2 and DAT in RCC patients demonstrated a high classification performance.

## Supporting information

**S1 Table. The differential genes expression dataset of ccRCC was obtained from the TCGA publish database via OncoDB.**
(XLSX)

**S2 Table. The differential genes expression dataset of pRCC was obtained from the TCGA publish database via OncoDB.**
(XLSX)

**S3 Table. The differential genes expression dataset of chRCC was obtained from the TCGA publish database via OncoDB.**
(XLSX)

**S4 Table. The GO and pathway enrichment of specific DEGs of ccRCC.**
(PDF)

**S5 Table. The GO and pathway enrichment of specific DEGs of pRCC.**
(PDF)

**S6 Table. The GO and pathway enrichment of specific DEGs of chRCC.**
(PDF)

**S7 Table. The Clinicopathological data of RCC patients.**
(XLSX)

## Acknowledgments

We thank the Department of Pathology at Srinagarind Hospital, Faculty of Medicine, Khon Kaen University, for their support. We also acknowledge Miss Yingpinyapat Kittirat (a post-doctoral researcher at the Cholangiocarcinoma Research Institute) for her suggestions for the immunohistochemistry experiments. Lastly, we thank Miss Sirirat Chain (the research assistant in our department) for her suggestions and help with statistical calculations.

## Author Contributions

**Conceptualization:** Chanita Panwoon, Sakkarn Sangkhamanon.

**Data curation:** Chanita Panwoon, Wunchana Seubwai.

**Formal analysis:** Chanita Panwoon, Wunchana Seubwai.

**Funding acquisition:** Sakkarn Sangkhamanon.

**Investigation:** Chanita Panwoon, Wunchana Seubwai, Sakkarn Sangkhamanon.

**Methodology:** Chanita Panwoon, Wunchana Seubwai, Malinee Thanee.

**Project administration:** Sakkarn Sangkhamanon.

**Resources:** Malinee Thanee, Sakkarn Sangkhamanon.

**Software:** Chanita Panwoon, Wunchana Seubwai.

**Supervision:** Wunchana Seubwai, Malinee Thanee, Sakkarn Sangkhamanon.

**Validation:** Chanita Panwoon, Wunchana Seubwai, Sakkarn Sangkhamanon.

**Visualization:** Chanita Panwoon.

**Writing – original draft:** Chanita Panwoon.

**Writing – review & editing:** Wunchana Seubwai, Malinee Thanee, Sakkarn Sangkhamanon.

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
