## [Decision Letter · Decision Letter 0]

15 Apr 2024

PONE-D-24-11162Identification of novel biomarkers to distinguish clear cell and non-clear cell renal cell carcinoma using bioinformatics and machine learningPLOS ONE

Dear Dr. Sangkhamanon,

Thank you for submitting your manuscript to PLOS ONE. After careful consideration, we feel that it has merit but does not fully meet PLOS ONE’s publication criteria as it currently stands. Therefore, we invite you to submit a revised version of the manuscript that addresses the points raised during the review process.

We look forward to receiving your revised manuscript.

Kind regards,

Giuseppe Lucarelli, M.D., Ph.D.

Academic Editor

PLOS ONE

“The study was supported by grants from the Fundamental Fund of Khon Kaen University 2566, under  the National Science, Research and Innovation Fund or NSRF, Thailand.”

4. For studies involving third-party data, we encourage authors to share any data specific to their analyses that they can legally distribute. PLOS recognizes, however, that authors may be using third-party data they do not have the rights to share. When third-party data cannot be publicly shared, authors must provide all information necessary for interested researchers to apply to gain access to the data. (https://journals.plos.org/plosone/s/data-availability#loc-acceptable-data-access-restrictions)

a) A description of the data set and the third-party source

b) If applicable, verification of permission to use the data set

c) Confirmation of whether the authors received any special privileges in accessing the data that other researchers would not have

d) All necessary contact information others would need to apply to gain access to the data

Reviewers' comments:

Reviewer's Responses to Questions

**Comments to the Author**

1. Is the manuscript technically sound, and do the data support the conclusions?

Reviewer #1: Partly

Reviewer #2: Yes

2. Has the statistical analysis been performed appropriately and rigorously? 

Reviewer #1: Yes

Reviewer #2: Yes

3. Have the authors made all data underlying the findings in their manuscript fully available?

Reviewer #1: Yes

Reviewer #2: Yes

4. Is the manuscript presented in an intelligible fashion and written in standard English?

Reviewer #1: Yes

Reviewer #2: Yes

5. Review Comments to the Author

Reviewer #1: In this study the authors identified NDUFA4L2 and DAT as novel biomarkers to differential diagnosis ccRCC from

non-ccRCC usign a bioinformatic approach.

I have some comments:

-Renal cell carcinoma (RCC) is essentially a metabolic disease characterized by a reprogramming of energetic metabolism (PMID: 36960789; PMID: 30983433, PMID: 36430837,PMID: 36310399). In particular the metabolic flux through glycolysis is partitioned (PMID: 29371925, PMID: 28933387, PMID: 25945836), and mitochondrial bioenergetics and OxPhox are impaired , as well as lipid metabolism (PMID: 30538212; PMID: 32861643, PMID: 29371925, PMID: 36430448, PMID: 38540735). In this scenario it has been shown by Lucarelli et al. that NDUFA4L2 is an important regulator of cell metabolism in ccRCC and regulates many biological characteristics of renal cancer stem cells (PMID: 37685983). These findings should be referenced and discussed.

-In addition, renal cell carcinoma is one of the most immune-infiltrated tumors (PMID: 31527133, PMID: 30738745; PMID: 27063186). Emerging evidence suggests that the activation of specific metabolic pathway have a role in regulating angiogenesis and inflammatory signatures (PMID: 32345771, PMID: 28359744). Features of the tumor microenvironment heavily affect disease biology and may affect responses to systemic therapy (PMID: 38003705; PMID: 37189689; PMID: 33265926; PMID: 36902242; PMID: 37373581). NDUFA4L2-induced metabolic alterations can modulate immune cell infiltration and regulate immunoflogosis. These processes should be explored and discussed.

- The entire study was based on data mining, the lack of independent validation with wet lab experiments using cell lines or clinical specimens weakens this study. The author should specify this limitation in the discussion

Reviewer #2: Renal cell carcinoma (RCC) is the most common type of urogenital cancer. . It comprises a diverse group of malignancies arising from the nephron. The World Health Organization (WHO) classifies RCC into different subtypes based on its morphologic, molecular, and genetic features. It’s important for the purpose of management and prognosis to differentiate clear cell RCC from nccRCC.

This article provides a comprehensive investigation into find biomarkers that can differentiate between clear cell RCC and non-clear cell RCC using advanced computational techniques. The study utilizes gene expression data from The Cancer Genome Atlas (TCGA) to identify differential expression genes (DEGs) specific to each RCC subtype, including papillary RCC (pRCC) and chromophobe RCC (chRCC).

The author identifies two new genes (NDUFA4L2 and DAT) as possible biomarkers in the differentiation of clear cell RCC and non-clear cell RCC.

There are some limitations. Firstly, the sample size is inadequate to generalize the results. Also, more detailed discussions on the biological functions and clinical implications of the identified biomarkers is needed.

I suggest adding the following scientific article links to the bibliography section for a more accurate representation of the references and the general topic of this study:

- 10.3390/ijms24054615 , an interesting review on new prospective in the management of renal cancer.

- 10.1177/17562872231164803, a literature review on the newest tools in kidney lesions evaluation

- 10.3390/diagnostics13132308, enhancing the use of AI in the urological field.

6. PLOS authors have the option to publish the peer review history of their article (what does this mean?). If published, this will include your full peer review and any attached files.

Reviewer #1: No

Reviewer #2: No

---

## [Author Response · Author response to Decision Letter 0]

20 May 2024

Editor comments

Answers: Thank you. We have adjusted the manuscript, including the file naming, according to the style requirements guideline.

2. Please note that PLOS ONE has specific guidelines on code sharing for submissions in which author-generated code underpins the findings in the manuscript. 

Answers: Thank you. We have uploaded code to GitHub with the following link below: https://github.com/CHaPAN6551/machine-learning-classification-algorithm

3. Please state what role the funders took in the study. 

Answers: Thank you. We have added “The funders had no role in study design, data collection, and analysis, manuscript preparation, or publication decisions” as a yellow highlight on Page no. 29, lines 439-440.

4. Third-party data

Answers: Thank you. We have added the information to the supporting information section. 

----------

Review Comments to the Author

Reviewer #1: 

In this study the authors identified NDUFA4L2 and DAT as novel biomarkers to differential diagnosis ccRCC from non-ccRCC using a bioinformatic approach.

I have some comments:

1) Renal cell carcinoma (RCC) is essentially a metabolic disease characterized by a reprogramming of energetic metabolism (PMID: 30983433, PMID: 36430837,PMID: 36310399). In particular the metabolic flux through glycolysis is partitioned (PMID: 29371925, PMID: 28933387, PMID: 25945836), and mitochondrial bioenergetics and OxPhox are impaired, as well as lipid metabolism (PMID: 30538212; PMID: 32861643, PMID: 29371925, PMID: 36430448, PMID: 38540735). In this scenario it has been shown by Lucarelli et al. that NDUFA4L2 is an important regulator of cell metabolism in ccRCC and regulates many biological characteristics of renal cancer stem cells (PMID: 37685983). These findings should be referenced and discussed.

Response to Reviewer: 

Thank you for your great suggestion. According to the reviewer's comment, We added some information in the discussion section (references no. 17-19) with the following sentences “RCC is suggested as a metabolic disease, with alterations in various pathways like energy metabolism, fatty acid metabolism, and amino acid processing (Schaeffeler et al., 2019; Wettersten, Aboud, Lara, & Weiss, 2017). Wettersten et al. suggest a reprogramming of sugar and fat usage, along with changes in the tricarboxylic acid cycle, often linked to mutations in the VHL gene (Wettersten et al., 2017). Lucarelli et al. reported that a high level of glucose-6-phosphate dehydrogenase (G6PD) because of increasing pentose phosphate pathway (PPP)-derived metabolites led to enhanced activity in both glycolysis and the PPP in ccRCC (Lucarelli et al., 2015). These findings suggest energy metabolic and glycolysis reprogramming is key in promoting RCC. Additionally, Schaeffeler and co-workers explored that RCC subtypes had different biological and metabolic profiles based on the correlation of the original region (Schaeffeler et al., 2019).”as yellow highlighted on Page no. 24, lines 319-327.

In addition, we added some discussion (reference no. 28) as the following sentences

- “The high expression of NDUFA4L2 regulates glucose and energy metabolism alteration in cancer cells, promoting increased glucose uptake, a shift towards glycolysis, and altered mitochondrial function (Lucarelli et al., 2018).” as yellow highlighted on Page no. 26, line no. 374-376

- “Conversely, NDUFA4L2 silencing also disrupts cellular processes, including inhibition of the autophagic machine, increased mitochondrial mass, and induced overproduction of ROS, especially in hypoxic conditions (Lucarelli et al., 2018).” as yellow highlighted on Page no. 26, line no. 381-383

2) In addition, renal cell carcinoma is one of the most immune-infiltrated tumors (PMID: 31527133, PMID: 30738745; PMID: 27063186). Emerging evidence suggests that the activation of specific metabolic pathways has a role in regulating angiogenesis and inflammatory signatures (PMID: 32345771, PMID: 28359744). Features of the tumor microenvironment heavily affect disease biology and may affect responses to systemic therapy (PMID: 38003705; PMID: 37189689; PMID: 33265926; PMID: 36902242; PMID: 37373581). NDUFA4L2-induced metabolic alterations can modulate immune cell infiltration and regulate immunoflogosis. These processes should be explored and discussed.

Response to Reviewer: 

Thank you for your kind suggestion. Regarding your advice, we added the text from references (references no. 32 and 33) in the discussion section with the following sentences: “ In particular, ccRCC stands out as a tumor with significant immune infiltration. This infiltration in tumors has been linked to the VHL tumor suppressor gene, triggering HIF-driven cellular responses, angiogenesis, and metabolic alterations (Vuong, Kotecha, Voss, & Hakimi, 2019). Furthermore, previous study has revealed that metabolic-associated genes like MUC1 can influence immunoflogosis within the ccRCC microenvironment by activating the classical pathway of the complement system and modulating immune infiltrates, thereby fostering an immune-silent microenvironment (Lucarelli et al., 2023). These findings displayed the hidden significance of NDUFA4L2 in potentially inducing modulation of immune cell infiltration and regulating immunoflogosis by controlling metabolic alterations in ccRCC. In summary, the evidence suggests that targeting NDUFA4L2 could be a potential immunotherapeutic target of ccRCC.” as yellow highlighted on Page no. 27, line no. 386-395

3) The entire study was based on data mining, the lack of independent validation with wet lab experiments using cell lines or clinical specimens weakens this study. The author should specify this limitation in the discussion.

Response to Reviewer: 

Thank you for your kind suggestion. As the reviewer mentioned, “Limitation of this study”. We added the limitation of this study in the discussion section as follows; “The larger sample size should further validate the expression of these genes. The investigation of functional study on important genes in in vitro model might be used to be a potential target for RCC treatment in the future.” as yellow highlighted on Page no. 28, line no. 421-423

----------

Reviewer #2: 

Renal cell carcinoma (RCC) is the most common type of urogenital cancer. It comprises a diverse group of malignancies arising from the nephron. The World Health Organization (WHO) classifies RCC into different subtypes based on its morphologic, molecular, and genetic features. It’s important for the purpose of management and prognosis to differentiate clear cell RCC from nccRCC.

This article provides a comprehensive investigation into find biomarkers that can differentiate between clear cell RCC and non-clear cell RCC using advanced computational techniques. The study utilizes gene expression data from The Cancer Genome Atlas (TCGA) to identify differential expression genes (DEGs) specific to each RCC subtype, including papillary RCC (pRCC) and chromophobe RCC (chRCC).

The author identifies two new genes (NDUFA4L2 and DAT) as possible biomarkers in the differentiation of clear cell RCC and non-clear cell RCC.

There are some limitations. Firstly, the sample size is inadequate to generalize the results. Also, more detailed discussions on the biological functions and clinical implications of the identified biomarkers are needed.

1) Clinical implications of the identified biomarkers are needed.

Response to Reviewer: 

Thank you for your kind suggestion. According to the reviewer's comment, we added some information in the discussion section on these two new genes (NDUFA4L2 and DAT) with the following sentences:

1.1) NDUFA4L2

- “This suggested NDUFA4L2 plays a role in resistant of chemotherapy of RCC.” as yellow highlighted on Page no. 27, line no. 385-386

- “Possibly, NDUFA4L2 might be a potential immunotherapeutic target in many cancers.” as yellow highlighted on Page no. 27, line no. 398-399

1.2) DAT

- “Previous studies suggest that DAT was expected to be a specific biomarker of prognosis or targeted therapy for ccRCC.” As a yellow highlighted on Page no. 27, line no. 405-406

- “DAT might be a serological diagnostic marker of gastric cancer.” As a yellow highlighted on Page no. 28, line no. 415-416

2) I suggest adding the following scientific article links to the bibliography section for a more accurate representation of the references and the general topic of this study:

2.1) 10.3390/ijms24054615, an interesting review on new prospective in the management of renal cancer.

Response to Reviewer: 

Thank you for your kind suggestion. According to the reviewer's comment, We added some information in the introduction section (reference no. 5) with the following sentences “Despite the promising approach of novel RCC management through radiogenomics, which offers the capability to characterize potentially malignant lesions in their genetic, epigenetic, and pathological heterogeneity via noninvasive methods such as advanced imaging study (Ferro, Musi, et al., 2023). The comprehensive understanding of the differences in transcriptomic and proteomic profiles between ccRCC and non-ccRCC remains unclear.” as yellow highlighted on Page no. 3, line no. 63-68

2.2) 10.1177/17562872231164803, a literature review on the newest tools in kidney lesions evaluation

2.3) 10.3390/diagnostics13132308, enhancing the use of AI in the urological field.

Response to Reviewer: 

Thank you for your thoughtful recommendation. Following your guidance, we have incorporated the content from references 20 and 21 into the discussion section. We considered merging these two points (2.2 and 2.3) with the following sentences: “In the era of precision medicine, artificial intelligence (AI) and machine learning (ML) have been applied using a multi-omics approach to enhance performance in detecting and differentiating urological tumors. Particularly successful were radiography studies for kidney and bladder lesions (Ferro, Crocetto, et al., 2023; Ferro, Falagario, et al., 2023). Integrating radiologic diagnosis with AI and ML is currently undergoing widespread development. However, there are limited studies in the pathology field. Thus, there is an opportunity to integrate transcriptomic analysis and ML that can be applied in pathology practice.” as yellow highlighted on Page no. 24, line no. 328-334

---

## [Editor Report · Decision Letter 1]

28 May 2024

Identification of novel biomarkers to distinguish clear cell and non-clear cell renal cell carcinoma using bioinformatics and machine learning

PONE-D-24-11162R1

Dear Dr. Sangkhamanon,

We’re pleased to inform you that your manuscript has been judged scientifically suitable for publication and will be formally accepted for publication once it meets all outstanding technical requirements.

Kind regards,

Giuseppe Lucarelli, M.D., Ph.D.

Academic Editor

PLOS ONE
---

## [Editor Report · Acceptance letter]

31 May 2024

PONE-D-24-11162R1 

PLOS ONE

Dear Dr. Sangkhamanon, 

I'm pleased to inform you that your manuscript has been deemed suitable for publication in PLOS ONE. Congratulations! Your manuscript is now being handed over to our production team.

Kind regards, 

on behalf of

Dr. Giuseppe Lucarelli 

Academic Editor

PLOS ONE